# Towards Context-Agnostic Learning Using Synthetic Data

**Charles Jin**
CSAIL
MIT
Cambridge, MA 02139
ccj@csail.mit.edu

**Martin Rinard**
CSAIL
MIT
Cambridge, MA 02139
rinard@csail.mit.edu

## Abstract

We propose a novel setting for learning, where the input domain is the image of a map defined on the product of two sets, one of which completely determines the labels. We derive a new risk bound for this setting that decomposes into a bias and an error term, and exhibits a surprisingly weak dependence on the true labels. Inspired by these results, we present an algorithm aimed at minimizing the bias term by exploiting the ability to sample from each set independently. We apply our setting to visual classification tasks, where our approach enables us to train classifiers on datasets that consist entirely of a single synthetic example of each class. On several standard benchmarks for real-world image classification, we achieve robust performance in the context-agnostic setting, with good generalization to real world domains, whereas training directly on real world data without our techniques yields classifiers that are brittle to perturbations of the background.

## 1 Introduction

We study the problem of learning to classifying images of known objects when placed in context, given only a single synthetic example of each object; our empirical evaluation considers the tasks of traffic sign and handwritten character recognition.

Our methods also enable us to explore the extent to which deep neural networks trained using real-world data to perform image classification can over-rely on signals from the backgrounds of images, even when no such information is necessary for classification. Intuitively, even though contextual clues may be required in some settings, given an input that *unambiguously* contains the object of interest in the foreground, we expect a robust classifier to be invariant to the rest of the image.

We present two main contributions. First, we introduce a formal setting for our study, where the input space is decomposed into object and context spaces, and the labels are independent of contexts when conditioned on the objects. We introduce the goal of learning a *context-agnostic* classifier, i.e., a classifier whose predictions are invariant under perturbations of the context. We derive a new risk bound for this setting that is *tight up to a factor of two* but also *independent of the true labels*, which holds so long as the classifier outperforms random guessing.

Second, we present a new technique for automatically generating data to train a deep neural network for image classification. The technique works by sampling independently from the object space, which contains the transformed views of the objects, and the context space, which is designed to include challenging backgrounds that make the resulting images difficult to classify. The hypothesis is that training the network to accurately classify objects against these challenging backgrounds should produce a trained network that robustly generalizes to accurately classify objects against the range of backgrounds that it may encounter when deployed in more natural settings.

35th Conference on Neural Information Processing Systems (NeurIPS 2021).

We empirically validate our methods by training deep neural networks for a variety of real-world image classification tasks using only a single synthetic example of each class, obtaining robust performance in the context-agnostic setting on natural data. Conversely, we find that classifiers trained without our techniques using only natural data achieve negligible accuracy even under relatively benign perturbations that leave a well-defined object in the foreground completely untouched. These results demonstrate the ability of our technique to train accurate and robust classifiers using only small amounts of high quality synthetic data, while also highlighting the need for future work in understanding the performance of deep learning systems in the context-agnostic setting when trained on natural data.

## 2    Related work

Domain shift refers to the problem that occurs when the training set (source domain) and test set (target domain) are drawn from different data distributions. In this setting, a classifier which performs well on the source domain may not generalize well to the target domain. A standard method for addressing this challenge is domain adaptation, which leverages a small amount of data from the target domain to adapt a function that is learned over the source domain [Blitzer et al., 2006]. Approaches can be further categorized as supervised, using small amounts of labelled data from the target domain [Donahue et al., 2014, Motiian et al., 2017b, Saenko et al., 2010, Tzeng et al., 2015]; unsupervised, typically requiring large amounts of unlabelled data from the target domain [Baktashmotlagh et al., 2013, Fernando et al., 2013, Ganin and Lempitsky, 2014, Gopalan et al., 2011]; or semi-supervised, using a mixture of both types of data [Gong et al., 2012, Pan et al., 2010, Yao et al., 2015].

In the context of learning from synthetic data, the domain shift that occurs between synthetic and real world data is known as the reality gap [Jakobi et al., 1995]. State-of-the-art rendering engines, such as those used for video games, can help narrow this gap by generating photorealistic data for training [Dosovitskiy et al., 2017, Johnson-Roberson et al., 2016, Qiu and Yuille, 2016]. Another technique, particularly prevalent in the robotics community, is known as domain randomization, where the synthetic training data is generated with more variability than expected in the actual test environment (e.g., extreme lighting conditions and camera angles), so that the additional robustness also transfers across the reality gap [Tobin et al., 2017, Tremblay et al., 2018]; in particular, Torres et al. [2019] apply domain randomization to traffic sign detection and find that arbitrary natural images suffice for the task. Another body of work exploits generative adversarial networks (GANs) [Goodfellow et al., 2014a] to generate synthetic domains [Hoffman et al., 2017, Liu et al., 2017, Shrivastava et al., 2016, Taigman et al., 2016, Tzeng et al., 2017]. In particular, Shetty et al. [2019] use a GAN trained to perform in-painting and replace extraneous objects in images as a data-augmentation technique to reduce the trained model's dependence on co-occurring classes.

A different paradigm for the low-data regime is few-shot learning. In contrast to domain adaptation, the goal of few-shot learning is to generalize to new classes given only a few examples, given the ability to train on large amounts of data containing related classes. Early approaches emphasized capturing knowledge in a Bayesian framework [Fe-Fei et al., 2003], which was later formulated as Bayesian program learning [Lake et al., 2015]. Another approach based on metric learning is to find a nonlinear embedding for objects where closeness in the geometry of the embedding generalizes to unseen classes [Koch, 2015, Snell et al., 2017, Sung et al., 2018, Vinyals et al., 2016]. Meta-learning approaches aim to extract higher level concepts which can be applied to learn new classes from a few examples [Finn et al., 2017, Munkhdalai and Yu, 2017, Nichol et al., 2018, Ravi and Larochelle, 2016]. A conceptually-related method that leverages synthetic training data is learning how to generate new data from a few examples of unseen classes; in contrast to our work, however, these methods still require a large number of samples to learn the synthesizer [Schwartz et al., 2018, Zhang et al., 2019]. Finally, some works combine domain adaptation with few-shot learning to learn under domain shift and limited samples (Motiian et al. [2017a]).

The main characteristic that differentiates our work from these approaches is that we are interested in learning classifiers that are *context-agnostic*, i.e., do not rely on background signals. As such, while we find our approach is applicable to many of the same tasks as the aforementioned works, our theoretical setting and objectives differ significantly. From a practical perspective, we demonstrate our techniques when *the entire training set consists solely of a single synthetic image of each class*, though our techniques can certainly be applied when more data is available; however the reverse does not hold, i.e., existing approaches for domain adaptation or few-shot learning cannot be applied to our

setting. Indeed, we consider this work to be complementary in that we are concerned with exploiting the additional structure that is inherent in certain domains, while the goal of domain adaptation and few-shot learning is to achieve good performance on a downstream target distribution given data from a related source distribution.

# 3 Context-agnostic learning

In this section, we introduce the formal setting of context-agnostic learning. We begin with some notation from the standard supervised learning setting. We are given an input space $\mathcal{X}$, an output space $\mathcal{Y}$, and a hypothesis space $\mathcal{H}$ of functions mapping $\mathcal{X}$ to $\mathcal{Y}$. A domain $P_D$ is a probability distribution over $(\mathcal{X}, \mathcal{Y})$. Given a target domain $P_T$ and a loss function $\ell$, the goal is to learn a classifier $h \in \mathcal{H}$ that minimizes the risk, i.e., the expected loss $R_{P_T}(h) := \mathbb{E}_{P_T}[\ell(h(x), y)]$. The training procedure is given as input a set of $n$ training samples $(x_1, y_1), ..., (x_n, y_n)$ drawn from a source domain $P_S$. The standard approach is empirical risk minimization, which takes the classifier that minimizes $R_{emp}(h) = \frac{1}{n} \sum_i \ell(h(x_i), y_i)$. This technique is known to converge to the optimal classifier over $P_S$ as the number of samples increases; furthermore, if $P_S$ is sufficiently "close" to $P_T$ (e.g., if $P_S = P_T$, as is the case when there is no domain adaptation), then such a classifier also achieves low risk in the target domain.

## 3.1 Formal setting

In general, we can frame the goal of classification as learning to extract reliable signals for the label $y$ from points $x \in \mathcal{X}$. This task is often complicated by the presence of noise or other spurious signals. However, for input spaces generated by physical processes, such signals are generally produced by distinct physical entities and can thus be thought of as independent signals that become mixed via an observation process. We aim to capture this additional structure in our setting.

Concretely, we have an object space $\mathcal{O}$, a context space $\mathcal{C}$, and an observation function $\gamma$ on $\mathcal{O} \times \mathcal{C}$. The input space $\mathcal{X}$ is defined as the image of $\gamma : \mathcal{O} \times \mathcal{C} \to \mathcal{X}$. We will assume that points in $\mathcal{O}$ are associated with a unique label in $\mathcal{Y}$, and points in $\mathcal{X}$ inherit labels from $\mathcal{O}$ via $\gamma$; in particular, we require that $\gamma$ be injective with respect to labels, i.e., $\gamma$ always maps objects with different labels in $\mathcal{O}$ to distinct points in $\mathcal{X}$.

In this work, we will consider the special case when $\mathcal{X} \subseteq \mathcal{C}$. Conceptually, the context space is an "ambient space" containing not only valid inputs, but also random noise or irrelevant classes; the input space is a subset of the context space for which there exists a well-defined label. For example, in our experiments we explore such a decomposition for the task of traffic sign recognition, where the object space $\mathcal{O}$ consists of traffic signs viewed from different angles, the context space $\mathcal{C}$ is unconstrained pixel space, and the input space $\mathcal{X}$ is the set of images that contain a traffic sign.

Recall that the standard objective of learning is to find a good classifier for an unknown subdomain $\mathcal{X}_{P_T} \subseteq \mathcal{X}$. We consider instead the task of learning a classifier on the entire input space $\mathcal{X}$. To sample from $\mathcal{X}$ we are given oracle access to the observation function and draw (labelled) samples from $\mathcal{O}$ and $\mathcal{C}$ independently. Clearly, if this problem is realizable, i.e., there exists $h^* \in \mathcal{H}$ for which $R_{\mathcal{X}}(h^*) = 0$, then we do not even need to know the target domain $P_T$, since

$$\mathcal{X}_{P_T} \subseteq \mathcal{X} \implies \left[ R_{\mathcal{X}}(h^*) = 0 \implies R_{\mathcal{P}_T}(h^*) = 0 \right] \tag{1}$$

Hence our new goal will be to learn a classifier over $\mathcal{X}$ which depends only on signals from $\mathcal{O}$; more precisely, we have the following definitions:

**Definition 3.1.** *A function $f$ on $\mathcal{X}$ is **context-agnostic** if*

$$\Pr[f \circ \gamma(o, c) = y] = \Pr[f \circ \gamma(o, c') = y] \qquad \forall c, c' \in \mathcal{C}, o \in \mathcal{O}, y \in \mathrm{Im}(f) \tag{2}$$

**Definition 3.2.** *The objective of **context-agnostic learning** is to find $h \in \mathcal{H}$ such that $h$ achieves the lowest risk of all context-agnostic classifiers.*

**Remark.** First, note that if the true label function $y^*$ is realizeable, then the lowest risk classifier is also context-agnostic. Second, we recover the standard supervised setting for the trivial context space $\mathcal{C} = \emptyset$. Conversely, our setting remains well-defined even in the trivial object space $\mathcal{O} = \{y_i\}_i$, the set of classes; however, this pushes all the complexity to the observation function $\gamma$, which may be

hard to define or intractable to compute. Finally, we do not preclude the existence of useful signals originating from the context for certain domains. For instance, a great deal of information can often be gleaned from the backgrounds of photos, e.g., stop signs are more often found in cities than on highways. Our theoretical setting avoids this issue by assuming realizability and uniqueness of labels; more practically, we argue that a "good" classifier should nonetheless recognize stop signs on the highway, and our experimental results provide evidence that over-reliance on such background signals leads to brittle classifiers.

## 3.2 A new risk bound for the context-agnostic setting

Our central tool in the context-agnostic setting is a new risk bound that decomposes into separate terms over the context and object spaces. We first develop a formal notion of contextual bias. For clarity we will assume a binary classification task and slightly abuse notation, denoting the classifier as $h$ instead of $h \circ \gamma$, i.e., $h : \mathcal{O} \times \mathcal{C} \to \{-1, 1\}$. We will denote the true label function as $y^*$.

**Definition 3.3.** *For an object $o \in \mathcal{O}$, the* expected classification $\bar{o}$ *and* object error $\hat{o}$ *are defined as*

$$\bar{o} := \mathbb{E}_{c \sim \mathcal{C}}[h(o, c)] \tag{3}$$

$$\hat{o} := |y^*(o) - \bar{o}| \tag{4}$$

**Definition 3.4.** *The* context bias $B(h, c)$ *of a classifier $h$ on the context $c$ is defined as*

$$sgn(B(h, c)) := sgn(\mathbb{E}_{o \sim \mathcal{O}}[h(o, c) - \bar{o}]) \tag{5}$$

$$||B(h, c)|| := \mathbb{E}_{o \sim \mathcal{O}}\big[\ell\big(h(o, c), \bar{o}\big)\big] \tag{6}$$

*where $\ell$ is the hinge loss $\ell(i, j) := \max(0, 1 - i * j)$.*

Intuitively, the sign of the bias corresponds to the label toward which the classifier is biased by a given context; the magnitude measures the strength of this bias. Clearly, the classifier is context-agnostic exactly when the bias is zero. We are now ready to state our main theoretical result, which gives an upper bound on the risk in terms of the context bias on $\mathcal{C}$ and object error over $\mathcal{O}$.

**Theorem 3.1.** *Let $h$ be a classifier with average bias $K$, and denote the true risk as $R(h)$. Then the risk can be lower bounded as*

$$K/2 \le R(h) \tag{7}$$

*with equality if and only if $R(h) = K = 0$. Furthermore, if the object error for all objects is bounded from above by $\alpha < 1$, then we may also upper bound the risk as*

$$R(h) \le K/(2 - \alpha) \tag{8}$$

*with equality if and only if all object errors equal $\alpha$.*

The proof is deferred to Appendix A. Since $\alpha < 1$, we can further upper bound $K/(2 - \alpha)$ as $K$, which yields the following concise corollary

$$K/2 \le R(h) < K \tag{9}$$

(under the same assumptions).

**Remark.** Both the learning objective and the standard definition of risk depend crucially on the true labels $y^*$; then given the setting of context-agnostic learning, one might be tempted to focus on minimizing some objective over the object space, which determines the labels exclusively. However from the theorem we see that measuring the context bias of a classifier gives a bound on the risk that is tight up to a constant factor of two *without access to any labels*. In fact, the dependence on the object space is fairly weak in that the labels only enter in via the assumption $\alpha < 1$, which is achieved exactly as soon as the classifier outperforms random guessing. Note that the error bound $\alpha$ and the bias bound $K$ are not independent; the conclusion is rather that the risk depends more strongly on a good estimate of the bias. In particular, $\alpha = 0$ if and only if $K = 0$ and $\alpha < 1$; observe also that when $\mathcal{C} = \emptyset$, then $K = 0$ holds trivially, but in this case we can identify $\mathcal{O}$ with $\mathcal{X}$, so the assumption that $\alpha < 1$ equivalently yields that the classifier is correct on all inputs.

---

**Algorithm 1:** Greedy Bias Correction

---

**Input:** Object space $\mathcal{O}$, context space $\mathcal{C}$, observation function $\gamma$, number of rounds $R$, resample probability $p$, classifier update subroutine `Fit`, binary classifier $h$

**Output:** Trained classifier $h$

*// initialize random context and label*
$c \sim \mathcal{C}$;
$y \sim \{-1, 1\}$;
**for** $r \leftarrow 1$ **to** $R$ **do**
    $o \sim \mathcal{O}(y)$; *// sample object*
    $x \leftarrow \gamma(o, c)$; *// observe object and context*
    $h \leftarrow \texttt{Fit}(h, x, y)$; *// perform classifier update*
    *// update context and label*
    $p' \leftarrow \texttt{Uniform}(0, 1)$;
    **if** $p' < p$ **then**
        *// resample random context and label*
        $c \sim \mathcal{C}$;
        $y \sim \{-1, 1\}$;
    **else**
        $c \leftarrow x$; *// previous image becomes new context*
        $y \leftarrow -y$; *// flip label*
    **end**
**end**

---

## 4 Context-agnostic learning of visual tasks using synthetic data

We now present a pair of algorithms for the setting of context-agnostic learning. The first algorithm is a generic approach to context agnostic learning that attempts to minimize the context bias in line with Theorem 3.1. The second algorithm is a specialization of our framework to the visual domain, where the observation function is given by superposition of objects over contexts. Both algorithms assume unlimited independent samples from $\mathcal{C}$ and $\mathcal{O}$, along with black-box access to the true labelling function $h^*$ and the observation function $\gamma$.

These assumptions allow us to learn $h^*$ simply by taking the number of samples to infinity. Unfortunately, learning a classifier on the entire input space $\mathcal{X}$ generally requires many more samples than learning a classifier on a smaller target domain $\mathcal{X}_{P_T}$. Thus we should aim to learn $h^*$ using as few samples as possible. Our main strategy we will be to exploit the *a priori* knowledge that the true label function $h^*$ is context-agnostic, and thus learn $h^*$ through the decomposed object and context spaces. To that end, note that while we only need $\max(|\mathcal{O}|, |\mathcal{C}|)$ samples to observe every object and context once, we need $|\mathcal{O}||\mathcal{C}|$ samples to observe every object in every context. Hence, the main challenge when the number of samples is low will be avoiding *spurious signals*, i.e., statistical correlations between context and objects (and by extension, labels) which are artifacts of the sampling process and do not generalize outside the training set. This intuition corresponds to the formal notion of contextual bias introduced in the previous section.

### 4.1 Greedy bias correction

The central idea behind Theorem 3.1 is leveraging the fact that labels depend only on objects to factor the risk into separate terms for object error and context bias. This factorization enables us to exploit our ability to sample independently from the object and context spaces. More specifically, we can use samples from $\mathcal{O}$ to minimize the object error, and samples from $\mathcal{C}$ to minimize the context bias. Since we only need $\alpha < 1$ (i.e., any performance that exceeds random guessing), we continue to draw objects randomly; however given an object $o$, we aim to observe it with the context for which the classifier has the strongest opposing bias. Intuitively, this allows the classifier to "correct" its bias and unlearn the spurious signals, thereby minimizing the bias and also the risk.

Adopting this approach without modification requires computing the bias of every context in $\mathcal{C}$. In most cases, however, even estimating a single bias may be prohibitively expensive. Thus, rather than

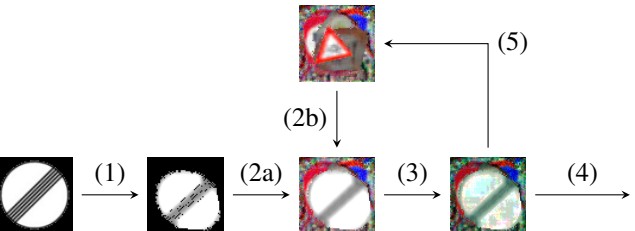

Figure 1: A graphical representation of the generative loop in Algorithm 2 using real training data. (1) Sample from object space. (2) Observe object and context. (3) Perform local refinement. (4) Add to training set. (5) Previous image becomes next context (resample from $\mathcal{C}$ with probability $p$).

solve for the maximum bias explicitly, we instead propose a heuristic for identifying contexts with large biases. Note that since $\mathcal{X} \subseteq \mathcal{C}$, a reasonable assumption is that the classifier learns a strong bias on recent training inputs when taken as contexts. This suggests a simple greedy approach for correcting biases by repurposing recent training inputs as contexts; we call this algorithm Greedy Bias Correction and present a description in Algorithm 1.

### 4.2   Learning visual tasks with synthetic data

We next introduce an instantiation of Greedy Bias Correction for learning visual tasks using synthetic data. We are given a function which takes a label $y$ and outputs a rendering of the corresponding class in a random pose without any background. The context is the background of the image, on which we place no restrictions. The observation function $\gamma$ superimposes an object over a background.

**Local refinement via robustness training**   We note that our observation function $\gamma$ is insufficient to capture the true range of possible inputs for a given object; for instance, we do not support occlusions. Because our ultimate goal will be to perform on data taken from a real-world context, we aim to capture this discrepancy using robustness training.[1] In particular, we assume that the image of $\gamma$ is an $\epsilon$-covering of $\mathcal{X}$, where a set $A$ is said to be an $\epsilon$-covering of another set $B$ iff for all points $b \in B$, there exists a point $a \in A$ such that $||a - b|| \leq \epsilon$. Then for a given sample $x$, we will instead add the point in the $\epsilon$-neighborhood of $x$ which maximizes the training loss, i.e., for a classifier $h$ and a sample $x = \gamma(o, c)$, we instead aim to identify

$$x' = \underset{x' \in N_\epsilon(x)}{\operatorname{argmax}} \ell(h(x'), y) \tag{10}$$

This formulation is often used to train models which are robust against local perturbations. An empirically effective method for finding approximations to $x'$ is known as Projected Gradient Descent (PGD) [Goodfellow et al., 2014b, Madry et al., 2017]. The algorithm can be summarized as

$$x_0 \leftarrow x + \delta \tag{11}$$
$$x_i \leftarrow \Pi_{N_\epsilon(x)}\big(x_{i-1} + \eta \cdot \operatorname{sgn}(\nabla_x \ell(h(x_{i-1}), y))\big), \quad i = 1, ..., n \tag{12}$$

where $\delta$ is a small amount of random noise, $\Pi$ is a projection back onto the $\epsilon$-ball, $\eta$ is the step size, and $n$ is the number of iterations. As is standard for robustness training, we take the $\epsilon$-ball with respect to the $\ell_\infty$-norm, defined as $||(x_1, ..., x_n)||_\infty = \max_i x_i$. Our choice of $\epsilon$ will depend on the task at hand, and we also use different $\epsilon$ for the portions of the image corresponding to the object and context.

Additionally, since we are no longer in a binary context, we sample a random permutation on labels instead of flipping the label deterministically. The full algorithm is presented as Algorithm 2 in Appendix B; Figure 1 provides a visualization of the key generative process, with images taken from a real step of training a deep neural network to perform classification of traffic signs.

From a practical standpoint, this algorithm makes concrete several benefits of our approach. First, rendering object classes, i.e. sampling from $\mathcal{O}$, is often relatively easy. In the case of two-dimensional

---

[1]Robustness training is more commonly referred to as adversarial training in the adversarial robustness community whence we borrow this technique. We use the nonstandard term to avoid confusion with the unrelated (generative) adversarial methods found in the few-shot learning literature.

Figure 2: Images from the training (top) and test (bottom) set for GTSRB (left) and MNIST (right).

Table 1: Performance of Algorithm 2 on various benchmarks, plus ablation studies.

| Approach | Picto → GTSRB | Digit → MNIST | Omnifont → Omniglot |
|---|---|---|---|
| baseline | 72.0 | 81.9 | 71.9 |
| + random-context | 72.1 | 88.3 | 69.8 |
|   + refinement-only | 86.4 | 89.7 | 90.8 |
| + bias-correction | 87.3 | 89.2 | 80.5 |
| **+ full** | **95.9** | **90.2** | **92.2** |

rigid body objects, this can be captured using standard data augmentation such as rotations, flips, and perspective distortions. Indeed, in this setting, our work can be viewed as a form of minimal one-shot learning, where the training data consists solely of a single unobstructed straight-on shot for each object class. Second, since we allow the context space $\mathcal{C}$ to be unconstrained, there is no requirement to perform realistic rendering of backgrounds, avoiding an additional layer of complexity.

Finally, because our approach is context agnostic, our classifiers are learned without any reference to target domains. In the formal setting, we assumed that the target domain was contained in the image of the observation function; however, synthetic images will always be subject to the reality gap. Our experiments suggest that our approach overcomes this barrier and successfully generalizes to natural images while training on synthetic data only.

## 5 Experiments

We evaluate our approach to learning visual tasks using synthetic data on three benchmarks for image recognition. Our training sets consist of a single synthetic image for each object class with no additional information about the target domain; Figure 2 shows examples of the training and test images from two of the datasets. On all three benchmarks, our models perform comparably with previous state-of-the-art results from related settings using few-shot learning and domain adaptation. Table 1 provides a summary of our results; comprehensive results and comparisons are compiled in Appendix D. Appendix C provides the full experimental setup and training details. Sample images from all datasets referenced below, including examples of rendered training data from the experiments and ablation studies, are shown in Appendix E.

### 5.1 GTSRB

For the traffic sign recognition task, our training set consists of a single, canonical pictogram of each class taken from the visualization software accompanying the target dataset, which we refer to as **Picto**. The target dataset is the German Traffic Sign Recognition Benchmark (**GTSRB**) [Stallkamp et al., 2012], which has 39,209 training and 12,630 test images of 43 classes of German traffic signs taken from the real world. We achieve 95.9% accuracy on the GTSRB test set training only on Picto, against a human baseline of 98.8%. A comprehensive comparison with existing approaches can be found in Appendix D, Table 2.

As a baseline, we also consider approaches using the **SynSign** [Moiseev et al., 2013] dataset of synthetic images designed to provide realistic training data for traffic sign recognition. The dataset comprises 100,000 synthetically generated images of signs from Sweden, Germany, and Belgium in a variety of poses, rendered against domain-appropriate backgrounds (e.g. trees, roads, sky) taken from real-world images. The dataset contains a superset of the GTSRB classes; as a result, Saito et al. [2017] report 79.2% accuracy by training directly on SynSign.

For domain adaptation, all approaches train on the full 100,000 images in SynSign plus part of the GTSRB training set. ATT [Saito et al., 2017] is the only method with better performance than ours, achieving 0.3% higher accuracy; however they use 31,367 unlabelled images from the GTSRB training set (in addition to SynSign). Methods using few-shot learning train on roughly half of the data (22 classes) from the GTSRB training set. The leading few-shot learning approach, VPE [Kim et al., 2019], adds a pictographic dataset similar to Picto, but achieves only 83.79% accuracy. In comparison, our training set consists of only 43 images, none of which are from GTSRB.

## 5.2 Handwritten character recognition

We consider two subtasks for handwritten character recognition. For the first subtask of classifying images of Arabic numerals, our training set, **Digit**, consists of a single example of each digit taken from a standard digital font. The target dataset, **MNIST** [LeCun], consists of 60,000 training and 10,000 test images of handwritten Arabic numerals in grayscale against a blank background. We achieve 90.2% accuracy on MNIST by training only on Digit, compared to human accuracy of 98%.

On MNIST, every approach using domain adaptation uses the full Street View House Numbers (**SVHN**) training set of 73,257 images of house numbers obtained from Google Street View [Netzer et al., 2011], plus varying amounts of data from MNIST. The domain shift problem faces a similar challenge as Digit, namely, handwriting exhibits different characteristics than house numbers fonts. Nevertheless, we note that SVHN contains far more examples of each digit. The only non-baseline approach to exceed our performance is CyCADA [Hoffman et al., 2017], which achieves 0.2% better performance by performing domain adaptation using 60,000 unlabelled images from the MNIST training set (in addition to training on SVHN). In contrast, we use only 10 images, none of which are from MNIST.

For the second subtask, we use the **Omniglot** [Lake et al., 2015] challenge, which consists of 1623 hand-written characters from 50 different alphabets, with 20 samples each. The samples were sourced online from 20 workers on Amazon's Mechanical Turk, who were asked to copy each character from a single font-based example using digital input (e.g., a mouse). We obtained the original representations (one per character) for our training images, which we call **OmniFont**. We achieve 92.2% on the 20-way Omniglot classification task training only on Omnifont, compared to human accuracy of 95.5%. Tables 3 and 4 in Appendix D compare our results on the handwritten character tasks with approaches using few-shot learning and domain adaptation.

Omniglot is often described as an MNIST-transpose, where the goal is learn handwriting rather than specific symbols, and is widely used as a benchmark for few-shot learning. We reproduce the most common split given in Lake et al. [2015], which uses a predefined set of 30 alphabets, with 19,280 images for training. Test performance is reported as an average over random subsets of $n = 5, 20$ unseen classes for the $n$-way task (given one labelled example). In comparison, for each test run, we retrain a model using only the corresponding $n$ images from OmniFont. As expected, our method finds 5-way classification easier than 20-way classification (95.8% vs 92.2%). In both cases, our performance lags behind the state-of-the-art for few-shot learning (>99%), though we emphasize that our experimental setup differs significantly in both the type and amount of training data used.

Finally, several approaches apply few-shot learning from Omniglot to MNIST, with the idea of transferring extracted features from human handwriting. We hypothesize that in comparison to Omniglot, where all the samples come from the same 20 subjects, MNIST may be particularly difficult for transfer one-shot learning, since any two examples will likely exhibit high "variance"; conversely, our approach benefits from using a canonical form which might be closer to the "mean" representation.

**Remark.** Handwritten characters and GTSRB present conceptually opposed challenges for learning: in GTSRB, the objects are rigid two-dimensional objects and backgrounds are complex settings in the natural world; in Omniglot and MNIST, backgrounds are uniform, but classes no longer have a strict specification and individual examples exhibit high variability. Thus, the main challenge of these tasks is learning how to generalize over the object class. Despite the inherent variation, a baseline model trained on Digit with plain data augmentation was able to achieve 81.9% accuracy on MNIST, exceeding many domain adaptation approaches and all the one-shot learning results; Omniglot is more difficult, with an Omnifont plus data augmentation baseline accuracy of 71.9%.

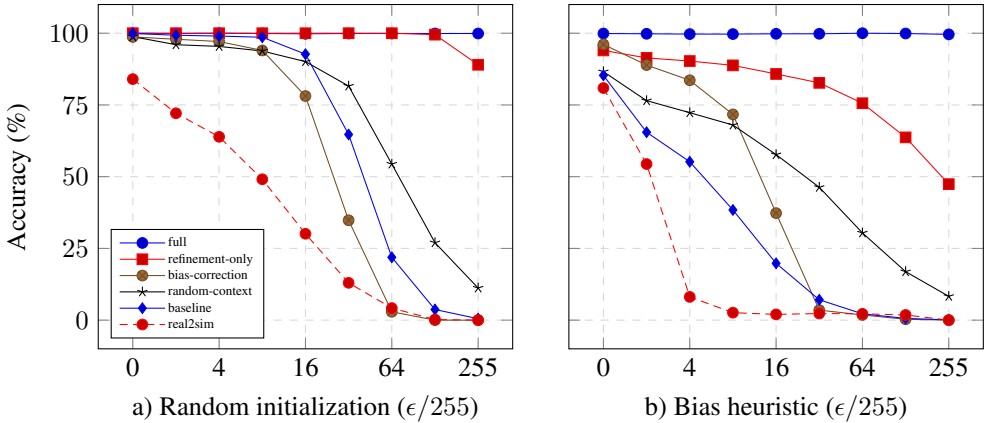

Figure 3: Context-agnostic performance on Picto using a `PGD` adversary on the background.

## 5.3 Ablation studies

We conduct two sets of ablation studies to better understand our approach to context-agnostic learning. The first study tests the individual components of our algorithm for their contributions to generalization over the real world dataset. All strategies employ the same data augmentation and use the following sampling procedures: **baseline** picks a fresh random background for each training point, and measures the performance of training on our synthetic dataset with plain data augmentation; **random-context** reuses random backgrounds as contexts; **bias-correction** reuses previous training images as contexts; **refinement-only** is the same as random-context with the addition of PGD-based refinement; **full** is the full algorithm as described in Algorithm 2. The results are in Table 1.

In all cases, we observe that both bias correction and local refinement contribute individually and jointly to the performance of our models. For GTSRB, a particularly interesting comparison is training on SynSign, a dataset designed to provide synthetic training data with realistic backgrounds for GTSRB, which yields 79.2% accuracy [Saito et al., 2017]. Though this is an improvement over our baseline of using random backgrounds at 72.0% accuracy, refinement-only and bias-correction achieve higher accuracy at 86.4% and 87.3%, respectively. Both methods leverage the background of training images to combat spurious signals, generating completely unrealistic backgrounds; this suggests that learning context-agnostic features is more effective than using realistic backgrounds.

The second study measures classification performance in a context-agnostic setting on the synthetic Picto dataset. By definition, the performance of a context-agnostic classifier should not degrade under perturbations of the background. We thus run an adaptive attack using a `PGD` adversary which fixes the foreground pixels, and ranges from fixed to unbounded on the background pixels, effectively searching the context space for a background that causes a misclassification on the given object. We also consider two initialization strategies for the `PGD` adversary: a standard random initialization, and initializing to the previous image, inspired by our bias heuristic.

We evaluate the same set of ablated strategies as before, plus a classifier trained directly on the GTSRB training set (discussed in the following section). Appendix E.2 contains samples of the generated images, and the results are plotted in Figure 3. Across all experiments, the models have worse (or very close) performance when using our bias heuristic for initialization. We believe this supports our usage of the bias heuristic for context-agnostic learning. Additionally, in the last column of Figure 3b, only our full method maintains passable accuracy, which suggests the gap between models is larger than performance on the GTSRB test set would indicate.

## 5.4 Comparison with training on real data

In this section, we compare a classifier trained using our method on only synthetic data with a model of the same architecture but trained directly on the GTSRB training set and achieving 98% performance on the GTSRB test set, which we refer to as **real2sim**. We first note that Figure 3 indicates the real2sim method seems to suffer from a "synthetic gap" even at $\epsilon = 0/255$, which is not entirely unexpected. However, in both settings, the performance of the real2sim model degrades very

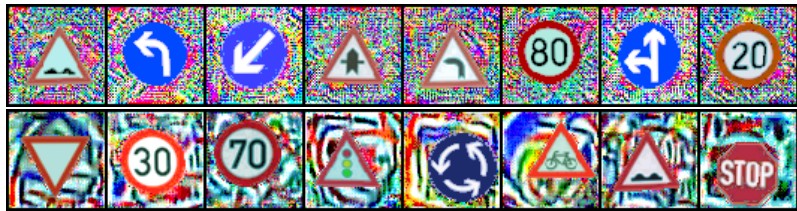

Figure 4: Test images from the second ablation study using the Picto dataset. Examples of test images generated using a PGD adversary initialized randomly (top) and with the bias heuristic (bottom) at $\epsilon = 255/255$. Note that only backgrounds are perturbed, while foregrounds remain unambiguous.

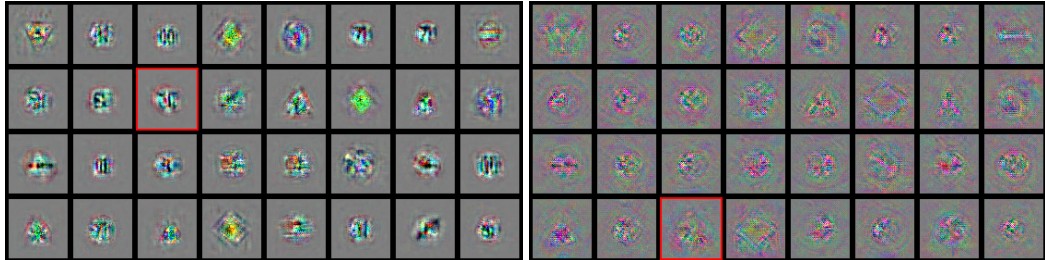

Figure 5: Guided Grad-CAM visualizations for the full (left) and real2sim (right) methods on the GTSRB test set. Misclassified images are marked with red boxes.

quickly as $\epsilon$ increases: the effect is most pronounced when the bias heuristic is used to initialize the PGD adversary, though in both cases the accuracy eventually drops to 0. We emphasize that all of the experiments leave the foreground objects completely unperturbed (and easily human-identifiable); Figure 4 presents examples of the generated test images.

To better understand the differences between a full classifier trained using our method and the real2sim classifier trained on real data, we used the Guided Grad-CAM method [Selvaraju et al., 2019], which localizes regions of fine-grained features that are important to the classifier's output. Figure 5 presents visualizations for the real2sim and full classifiers on images from the GTSRB test set. We see that the real2sim classifier (on the right) trained on real images of traffic signs has a more diffuse activation map with no clear interpretation, whereas the classifier trained using our method using purely synthetic data (on the left) is more focused, with more semantically-aligned features. Our results thus suggest that classifiers trained on natural images can become over-reliant on contextual signals, leading to surprisingly brittle behavior even given unambiguous foregrounds.

## 6 Conclusion

We introduce the task of context-agnostic learning, a theoretical setting for learning models whose predictions are independent of background signals. Leveraging the ability to sample objects and contexts independently, we propose an approach to context-agnostic learning by minimizing a formally defined notion of context bias. Our algorithm has a natural interpretation for training classifiers on vision-based tasks using synthetic data, with the distinct advantage that we do not need to model the background. We evaluate our methods on several real-world domains; our results suggest that our approach succeeds in learning context-agnostic classifiers that generalize to natural images using only a single synthetic image of each class, while training with natural images can lead to brittleness in the context-agnostic setting. Our performance is competitive with existing methods for learning when data is limited, while using significantly less data. More broadly, the ability to learn from single synthetic examples of each class also affords fine-grained control over the data used to train our models, allowing us to sidestep issues of data provenance and integrity entirely.

## Acknowledgments and Disclosure of Funding

We gratefully acknowledge support from DARPA Grant HR001120C0015. The views expressed are those of the authors and do not reflect the official policy or position of the Department of Defense or the U.S. Government.

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
