# A  Proofs

*Proof of Theorem 3.1.* Recall that we denote the true label function as $y^*$. First, we show the upper bound. By the assumption that $\alpha < 1$, we have that for all $o \in \mathcal{O}$, the signs of the expected classification $\bar{o}$ and correct classification $y^*(o)$ match, so that $\alpha \geq \hat{o} = |y^*(o) - \bar{o}| = 1 - |\bar{o}|$. Then for all $o \in \mathcal{O}$,

$$\ell(\bar{o}, y^*(o)) = 1 - \bar{o}y^*(o) \tag{13}$$

$$= 1 - |\bar{o}| \tag{14}$$

$$= \frac{1 - \bar{o}\bar{o}}{1 + |\bar{o}|} \tag{15}$$

$$= \frac{\ell(\bar{o}, \bar{o})}{1 + |\bar{o}|} \tag{16}$$

$$\leq \frac{\ell(\bar{o}, \bar{o})}{2 - \alpha}, \tag{17}$$

with equality if and only if $\alpha = \hat{o}$. Now to bound the risk, we can write,

$$R(h) := \mathbb{E}_{o \sim O, c \sim C}[\ell(h(o, c), y^*(o))] \tag{18}$$

$$= \int_{\mathcal{C}} \int_{\mathcal{O}} \ell(h(o, c), y^*(o)) \tag{19}$$

$$= \int_{\mathcal{O}} \ell(\bar{o}, y^*(o)) \tag{20}$$

$$\leq \int_{\mathcal{O}} \frac{\ell(\bar{o}, \bar{o})}{2 - \alpha} \tag{21}$$

$$= \frac{1}{2 - \alpha} \int_{\mathcal{O}} \int_{\mathcal{C}} \ell(h(o, c), \bar{o}) \tag{22}$$

$$= \frac{1}{2 - \alpha} \int_{\mathcal{C}} ||B(h, c)|| \tag{23}$$

$$= \frac{K}{2 - \alpha}, \tag{24}$$

where all double integrals are exchangeable by Fubini's theorem. It also follows that equality for the upper bound holds if and only if $\alpha = \hat{o}$ for all $o$.

For the lower bound, notice that for all $o \in \mathcal{O}$, we have $|\bar{o}| \leq 1$, so

$$\ell(\bar{o}, y^*(o)) \geq 1 - |\bar{o}| \tag{25}$$

$$= \frac{\ell(\bar{o}, \bar{o})}{1 + |\bar{o}|} \tag{26}$$

$$\geq \frac{\ell(\bar{o}, \bar{o})}{2}, \tag{27}$$

where line 25 is an equality if and only if the signs of the expected and correct classifications match, and line 27 is an equality if and only if $|\bar{o}| = 1$; together, these two conditions imply that the object is correctly classified over all contexts. Then a similar computation as for the upper bound shows that

$$R(h) \geq \int_{\mathcal{O}} \frac{\ell(\bar{o}, \bar{o})}{2} \tag{28}$$

$$= \frac{K}{2}, \tag{29}$$

where now equality holds if and only if all objects are correctly classified over all contexts, i.e., $R(h) = 0$. $\qquad\square$

# B   Greedy Bias Correction for visual tasks

---

**Algorithm 2:** Visual Learning Using Context-Agnostic Synthetic Data

---

**Input:** Object space $\mathcal{O}$, context space $\mathcal{C}$, random permutations $\Pi$, observation function $\gamma$,
  number of rounds $R$, batch size $B$, number of classes $N$, resample probability $p$,
  classifier update subroutine `Fit`, projected gradient descent subroutine `PGD`,
  classifier $h$

**Output:** Trained classifier $h$

**for** $r \leftarrow 1$ **to** $R$ **do**
    *// initialize empty training batch and random contexts*
    $X \leftarrow \emptyset$;
    **for** $n \leftarrow 1$ **to** $N$ **do**
      $c_n \sim \mathcal{C}$;
    **end**
    **for** $b \leftarrow 1$ **to** $B$ **do**
      *// sample random permutation*
      $\pi \sim \Pi(N)$;
      *// generate new training data*
      **for** $n \leftarrow 1$ **to** $N$ **do**
        $o \sim \mathcal{O}(n)$; *// sample object for class*
        $x \leftarrow \gamma(o, c_{\pi(n)})$; *// observe object and random (permuted) context*
        $x' \leftarrow \texttt{PGD}(h, x)$; *// perform local refinement*
        $X \leftarrow X \cup \{(x', y)\}$; *// add to training set*
        $c_n \leftarrow x'$; *// previous sample becomes next context*
      **end**
      *// resample contexts*
      **for** $n \leftarrow 1$ **to** $N$ **do**
        $p' \leftarrow \texttt{Uniform}(0, 1)$;
        **if** $p' < p$ **then**
          $c_n \sim \mathcal{C}$;
        **end**
      **end**
    **end**
    *// perform classifier update*
    $h \leftarrow \texttt{Fit}(h, X)$;
**end**

---

# C   Experimental setup

We used PyTorch 1.5.0 [Paszke et al., 2019], OpenCV 4.2.0 [Bradski, 2000], and scikit-image 0.17.2 [van der Walt et al., 2014] for all experiments. In setting the number of epochs, we did not observe any significant degradation or improvements in performance when training for longer. We use fewer epochs in the case of Omniglot due to computational constraints, as the model is retrained for each test split.

For GTSRB, we use a 5-layer convolutional neural network adapted from the official PyTorch tutorials. To train with Picto, the data augmentation consists of PyTorch transforms RandomAffine(5, translate=(.15, .15), scale=(0.65, 1.05), shear=5), RandomPerspective(0.5, p=1); ColorJitter(brightness=.8, contrast=.8, saturation=.8, hue=.05); OpenCV box blur with a random kernel size between 1 and 6 in both dimensions (independently sampled, so not necessarily square); and a random exposure adjustment by adjusting all pixels by the same random amount between –30% and 50%. For refinement, we used step sizes of $\alpha = 2/255$ with 8 steps and an epsilon of $\epsilon = 4/255$ for the foreground only. For the observation function, we superimpose the segmented foreground of the transformed pictographic sign over the context. We train for 300 epochs using the Adam optimizer (learning rate 1e-4, weight decay 1e-4), with 5 examples of each class per batch and 20 batches per epoch. We report results for the model that achieves the best performance on the training set, checking every 5 epochs.

For MNIST, we use the two-layer convolutional neural network from the official PyTorch examples for MNIST, with Dropout regularization replaced with pre-activation BatchNorm. To train with Digit, the data augmentation consists of PyTorch transforms RandomAffine(15, translate=(.15, .15), scale=(0.75, 1.05), shear=40), RandomPerspective(0.5, p=1); OpenCV box blur with a random kernel size between 1 and 6 in both dimensions (independently sampled, so not necessarily square); then set the foreground to all pixels with value greater than 0.2. For refinement, we used step sizes of $\alpha = 1.6/255$ with 8 iterations and no projection ($\epsilon = \infty$). For the observation function, we blend the object with the context at a 2:1 ratio; this ensures that inputs have a well-defined ground truth label. We train for 300 epochs using the Adam optimizer (learning rate 1e-4, weight decay 1e-4), with 5 examples of each class per batch and 20 batches per epoch. We report results for the model that achieves the best performance on the training set, checking every 5 epochs.

For Omniglot, we use the pre-activation variant of ResNet18 [He et al., 2015]. To train with Omnifont, we first preprocess with scikit-learn skeletonize and dilation to standardize stroke widths. Data augmentation consists of PyTorch transforms RandomAffine(15, translate=(.15, .15), scale=(0.75, 1.1), shear=20), RandomPerspective(0.25, p=1); OpenCV box blur with a random kernel size between 1 and 3 in both dimensions (independently sampled, so not necessarily square); then resize the images to 28 by 28. For refinement, we used step sizes of $\alpha = 1.6/255$ with 8 iterations and no projection ($\epsilon = \infty$). For the observation function, we blend the object with the context at a 2:1 ratio; this ensures that inputs have a well-defined ground truth label. For the $n$-way classification task, we randomly sample $n$ characters from the Omniglot test set, and use the corresponding characters from the Omnifont dataset as our training set. We then train a fresh model for 150 epochs using the Adam optimizer (learning rate 1e-4, weight decay 1e-4), and report performance on the all $20n$ images in the Omniglot test set, averaged over 20 runs (10 runs for the ablation studies).

For the GradCAM visualizations, we use the public grad-cam package [Gildenblat and contributors, 2021] from the Python Package Index (PyPI), with the target_layer set to the last LeakyReLU layer in the encoder, and both aug_smooth and eigen_smooth set to true.

# D Full experimental results

We compare a model trained using our methods with previous state-of-the-art results from related settings using few-shot learning and domain adaptation on GTSRB (Table 2), MNIST (Table 3), and Omniglot (Table 4). When multiple experiments are reported for the same approach, we compare against both the most accurate result as well as the result using the least amount of target data. We distinguish between labelled (**L**) and unlabelled (**UL**) data; experiments for which the training data is not known are marked (**?**).

Table 2: GTSRB results.

| Approach | Method | Training Data Source | Training Data Target | Accuracy (%) |
|---|---|---|---|---|
| Baselines | Source Only (Saito et al. [2017]) | SynSign | | 79.2 |
| | Human (Stallkamp et al. [2012]) | | | 98.8 |
| | Target Only (Ganin et al. [2016]) | | All L | 99.8 |
| Few-Shot Learning | VPE (Kim et al. [2019])[§] | Picto[*] | 22 classes L | 83.8 |
| | MatchNet (Vinyals et al. [2016])[§] | | 22 classes L | 53.3 |
| | QuadNet (Kim et al. [2018])[§†] | | 22 classes L | 45.3 |
| Domain Adaptation | DSN (Bousmalis et al. [2016]) | SynSign | 1280 UL | 93.0 |
| | ML (Schoenauer-Sebag et al. [2019])[§] | SynSign | 22 classes L | 89.1 |
| | MADA (Pei et al. [2018])[§‡] | SynSign | 22 classes L | 84.8 |
| | DANN (Ganin et al. [2016]) | SynSign | 31367 UL | 88.7 |
| | ATT (Saito et al. [2017]) | SynSign | 31367 UL | **96.2** |
| Context Agnostic | baseline | Picto | 0 | 72.0 |
| | + random-context | Picto | 0 | 72.1 |
| | + refinement-only | Picto | 0 | 86.4 |
| | + bias-correction | Picto | 0 | 87.3 |
| | + full | Picto | 0 | **95.9** |

[§]Test accuracy on remaining 21 unseen classes.

[*]Kim et al. [2019] use a pictographic dataset similar to Picto.

[†]Reported in Kim et al. [2019].

[‡]Reported in Schoenauer-Sebag et al. [2019].

Table 3: MNIST results.

| Approach | Method | Training Data Source | Training Data Target | Accuracy (%) |
|---|---|---|---|---|
| Baselines | Human (Netzer et al. [2011]) | | | 98.0 |
| | Target Only (Tzeng et al. [2017]) | | All L | 99.2 |
| Few-Shot Learning | FADA (Motiian et al. [2017a]) | SVHN | 1 L / class | 72.8 |
| | + more data | SVHN | 7 L / class | 87.2 |
| | SiamNet (Koch [2015]) | Omniglot | 1 L / class | 70.3 |
| | MatchNet (Vinyals et al. [2016]) | Omniglot | 1 L / class | 72.0 |
| | APL (Ramalho and Garnelo [2019]) | Omniglot | 1 L / class | 61.0 |
| | + more data | Omniglot | ?[‡] | 86.0 |
| Domain Adaptation | DSN (Bousmalis et al. [2016]) | SVHN | 1000 UL | 82.7 |
| | DRCN (Ghifary et al. [2016]) | SVHN | ? | 81.9 |
| | DANN (Ganin et al. [2016]) | SVHN | ? | 73.9 |
| | ATT (Saito et al. [2017]) | SVHN | ? L + 1000 UL | 86.0 |
| | ADDA (Tzeng et al. [2017]) | SVHN | 60,000 UL | 76.0 |
| | CyCADA (Hoffman et al. [2017]) | SVHN | 60,000 UL | **90.4** |
| Context Agnostic | baseline | Digit | 0 | 81.9 |
| | + random-context | Digit | 0 | 88.3 |
| | + refinement-only | Digit | 0 | 89.7 |
| | + bias-correction | Digit | 0 | 89.2 |
| | + full | Digit | 0 | **90.2** |

[‡]Cumulative accuracy from adapting over the test set.

Table 4: Omniglot results for one-shot classification.[‡]

| Approach | Method | Training Data | Accuracy (%) 5-way | Accuracy (%) 20-way |
|---|---|---|---|---|
| Baselines | Human (Lake et al. [2015]) | | | 95.5 |
| Few-Shot Learning | MANN (Santoro et al. [2016]) | Omniglot | 82.2 | |
| | SiamNet (Koch [2015]) | Omniglot | 96.7[§] | 92.0 |
| | MatchNet (Vinyals et al. [2016]) | Omniglot | 98.1 | 93.8 |
| | PN (Snell et al. [2017]) | Omniglot | 98.8 | 96.0 |
| | BPL (Lake et al. [2015]) | Omniglot | | 96.7 |
| | APL (Ramalho and Garnelo [2019]) | Omniglot | 97.9 | 97.2 |
| | RN (Sung et al. [2018]) | Omniglot | 99.6 | 97.6 |
| | MAML++ (Antoniou et al. [2018]) | Omniglot | 99.5 | 97.7 |
| | TapNet (Yoon et al. [2019]) | Omniglot | | 98.1 |
| | GCR (Li et al. [2019]) | Omniglot | **99.7** | **99.6** |
| Context Agnostic | baseline | Omnifont | | 71.9 |
| | + random-context | Omnifont | | 69.8 |
| | + refinement-only | Omnifont | | 90.8 |
| | + bias-correction | Omnifont | | 80.5 |
| | + full | Omnifont | **95.8** | 92.2 |

[‡]The exact set up of the one-shot classification task often varies between authors. We believe the broad performance numbers are still useful for contextualizing our approach, and refer the reader to the original works for details.

[§]As reported in Vinyals et al. [2016]

# E   Training and test set visualizations

## E.1   Datasets

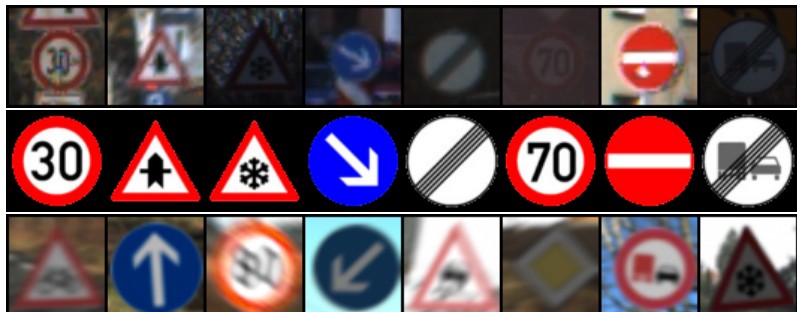

Figure 6: From top to bottom: samples from the GTSRB test set, Picto dataset, and SynSign dataset.

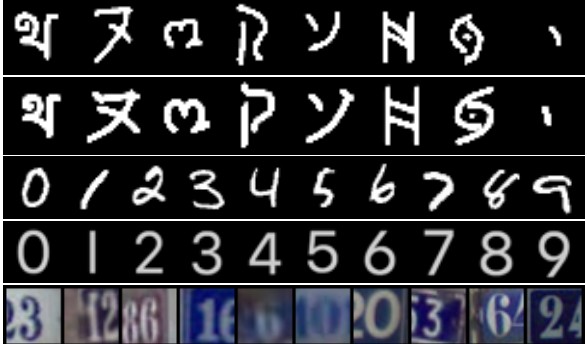

Figure 7: From top to bottom: samples from the Omniglot test set, Omnifot dataset, MNIST test set, Digit dataset, and SVHN training set.

## E.2 Ablation studies

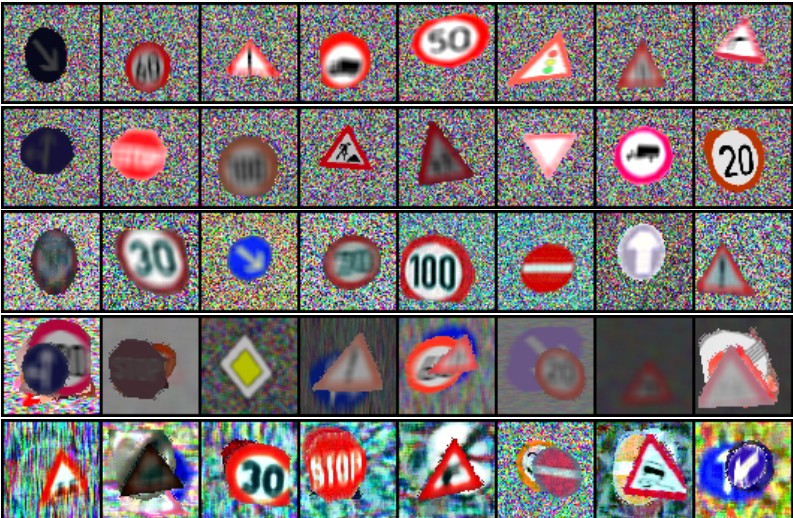

Figure 8: Training images from the first ablation study using Picto dataset. From top to bottom: baseline, random-context, refinement-only, bias-correction, full.

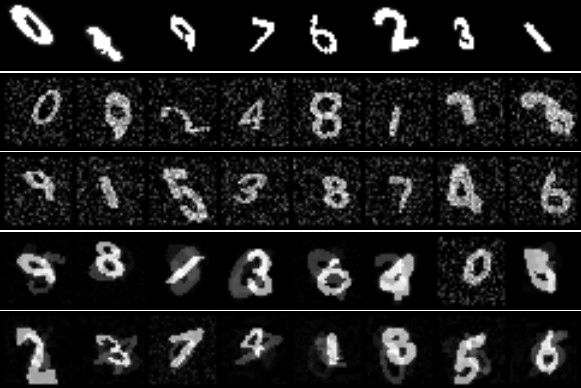

Figure 9: Training images from the first ablation study for the Digit dataset. From top to bottom: baseline, random-context, refinement-only, bias-correction, full.

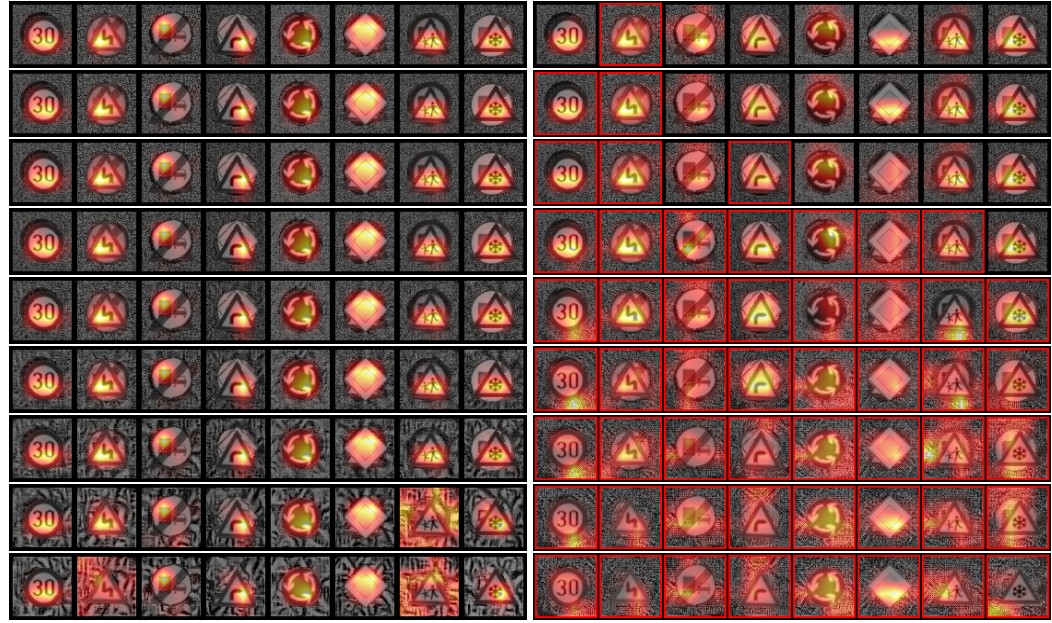

Figure 10: Grad-CAM visualizations for the PGD background perturbation ablation studies (initialized using the bias heuristic). Full (left) and real2sim (right) methods as perturbations increase over $\epsilon = 0, 2, 4, 8, 16, 32, 64, 128, 255$ (in order from top to bottom). Regions in yellow are more important to the classifier output. Misclassified images are marked with red boxes.

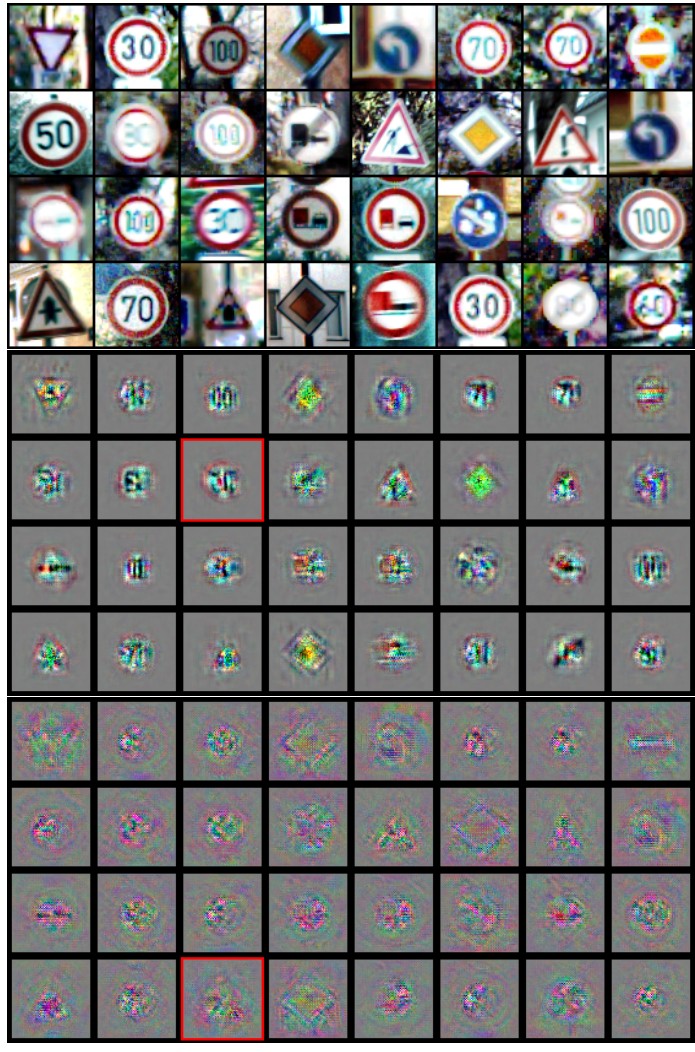

Figure 11: Guided Grad-CAM visualizations for the full (middle) and real2sim (bottom) methods on the GTSRB test set (original images on top). Regions with color visualize the fine-grained features that contribute to the classifier output. Misclassified images are marked with red boxes.