# OpenReview forum: "Towards Context-Agnostic Learning Using Synthetic Data"
_NeurIPS.cc/2021/Conference — NeurIPS 2021 Poster_

### Official Review · Reviewer_XVQW · 2021-07-16

**Rating:** 7
**Confidence:** 3

**Summary:**

The authors mathematically show that the risk of a classifier is bounded by K / (2 - \alpha) if the object error of the classifier is bounded by \alpha. This means that decreasing K (context bias) can improve the performance. Inspired by this and by assuming that we have access to a function that generates an image based on object O and context C, the authors propose an algorithm that is context-agnostic. Their algorithm updates the model based on generated image and then uses this new image to generate the context for the next image. This way, the classifier is enforced to ignore the context. They evaluate their method on three different datasets.

**Limitations And Societal Impact:**

Maybe the authors can add a paragraph and describe what could be the good and bad impacts of having classifiers or object detectors that generalize to context shifts. This could be beneficial for many research areas like robotics, self-driving cars, etc. It would be good to also study negative impacts and limitations. For example, the authors can describe what are the challenges when we face with more difficult datasets and what do they think are the way to overcome those challenges.

**Main Review:**

The paper is an innovative work, and the mathematical proof of showing that classifier accuracy is bounded by context bias is new and very interesting. The Proposed method on object recognition inspired by this mathematical proof is also novel. The authors study the related work and cover a reasonable number of related studies. They discuss few-shot learning and domain shift. I would suggest that the authors try to cite a couple of more related works that look at both domain shift and few-shot learning. Currently, they only cite one related work[1], but I suspect that there is just one work that studies both domain shift and few-shot learning. The paper is technically sound, and claims are supported in experimental results to a good degree. However, The authors could do experiments on more datasets like the Office dataset that is used by the related work they mention[1]. It is interesting to see if it is possible to extend this work to other objects such as chairs or monitors? Because it is not as easy to apply transformations on these objects as it is on characters of MNIST or Omniglot or traffic signs. For example, the authors apply blur or threshold transformation on images. However, chairs have different colours/shapes. Can the proposed algorithm learn that the colour belongs to the context? In addition, the authors can visualize the network activation on a given image from MNIST, GTSRB or Omniglot. This visualization can show how the network's decision comes from the pixels that belong to the object rather than pixels that belong to the context. I would increase my rating if these experiments or visualizations were added to the paper. Overall, I think the paper has substantial contributions and recommend a score of 6. In addition, The paper is well-structured and easy to follow. The idea is interesting and can be useful in different scenarios. This paper opens some interesting directions for learning classifiers and object detectors such that we can generalize better to domain shifts. This can help us advance the state-of-the-art by focusing on how we can recognize objects or train context-free classifiers on larger datasets. By itself, this work still is useful by some practitioners. For example, self-driving cars need to have a very robust context-free classifier to recognize road signs.
Finally, in line 100: Can you elaborate more? How can we generalize this to a case when the image \gamma is a subdomain of X?

=======Post Rebuttal=========
I found the authors' responses convincing and strongly suggest modifying the manuscript to include the new suggestions by the reviewers. As a result, I will increase my score to 7 since I think the community can benefit from this work.

[1]: Motiian S, Jones Q, Iranmanesh S, Doretto G. Few-Shot Adversarial Domain Adaptation. Advances in Neural Information Processing Systems.

**Time Spent Reviewing:**

8

---

> ### Author Response · Authors · 2021-08-10
> **Author response**
>
> We thank the reviewer for reading our work and the supportive comments! We have responded to some selected points below:
>
> > However, The authors could do experiments on more datasets like the Office dataset that is used by the related work they mention[1]. It is interesting to see if it is possible to extend this work to other objects such as chairs or monitors? Because it is not as easy to apply transformations on these objects as it is on characters of MNIST or Omniglot or traffic signs.
>
> The main criterion for selecting our datasets was a clean separation between the object and context spaces. We believe this is necessary for producing experimental results that can support our proposed theoretical framework. For instance, if altering the context also introduces visual artifacts, then any differences in performance could be attributed to either context dependency or an unintended out-of-distribution shift.
>
> > For example, the authors apply blur or threshold transformation on images. However, chairs have different colours/shapes. Can the proposed algorithm learn that the colour belongs to the context?
>
> This is an interesting question. Our theory and results suggest that this is possible, but one must explicitly include the color in the context when sampling. In other words, one must also be able to set the color of the chair in the training data. However, this is not possible with the Office dataset, which makes it unsuitable for the scope of this work.
>
> > In addition, the authors can visualize the network activation on a given image from MNIST, GTSRB or Omniglot. This visualization can show how the network's decision comes from the pixels that belong to the object rather than pixels that belong to the context. I would increase my rating if these experiments or visualizations were added to the paper.
>
> Thanks for the great suggestion. We think visualizing the attention map makes a lot of sense, and will include these results in our revision. Our preliminary experiments indicate that the networks trained using our methods focus on the foreground pixels, while the networks trained using standard methods have more diffuse attention maps and are easily fooled into depending on the background pixels.
>
> > Finally, in line 100: Can you elaborate more? How can we generalize this to a case when the image \gamma is a subdomain of X?
>
> We will clarify this in our revision. We can only provide performance guarantees for inputs in the image of $\gamma$. However, there may be other inputs (e.g., pure white noise), which the observation function cannot produce, but are also irrelevant in terms of performance.

---

### Official Review · Reviewer_PrQD · 2021-07-18

**Rating:** 5
**Confidence:** 4

**Summary:**

The submission proposes a motified setting where the task is context-agnostic learning and proposes an algorithm to solve the problem while assuming the ability to sample objects and contexts independently. The method achieves high accuracy on two synthetic visual tasks, digits and traffic sign classification, when a model is trained using one sample per class from the source domain and tested on an unseen target domain.



**Main Review:**

My main concern is experiments.  Although the method shows promising results under this specific setting where only one image is needed, it is hard to conclude that the proposed method will generalize other settings, such as when there are more classes, more images per classes (since one image per class is a very extreme setting), and other more complicated images (currently seems the images are simple and objects always sits in the middle).

I think evaluation on additional datasets would be necessary. It would be interesting to know if the proposed method could use more than one sample in order to make the comparison with SoA methods fair, while assessing performance in comparative terms with Sota, to give value to the method also in relation to performance.

Synthetic images are small iconic images with objects in the center. Although the method shows promising results under this specific setting, more datasets can have stronger support. It would also be more solid if there's more comparison with the generation based method  as described in related works A conceptually-related method that leverages synthetic training data is learning how to generate new
data from a few examples of unseen classes...

Also, I don't see many improvements compared to the last submission.


**Time Spent Reviewing:**

1

---

> ### Author Response · Authors · 2021-08-10
> **Author response**
>
> We thank the author for the comments. We see our contributions as two-fold: first, we introduce a formal setting for context agnostic learning, and prove a new risk bound. We then leverage this risk bound to provide a training algorithm for this setting, and demonstrate empirically that our approach achieves good context agnostic performance, while training directly on data does not.
>
> As such, while we do think our performance is a major contribution (as far as we know, we are the first to demonstrate the possibility of learning to classify real-world images given only a single image of each class), we think it is also important that our experiments are more closely aligned with the theory presented. In particular, we find also that a model trained directly on the GTSRB (traffic sign) training set achieves 0 performance in the context-agnostic setting. This experiment is only possible given the ability to sample the foreground independently of the background. In this case we do not think that larger datasets would be an improvement over the existing datasets.

---

### Official Review · Reviewer_gTVT · 2021-07-18

**Rating:** 6
**Confidence:** 4

**Summary:**

This paper proposes a data augmentation scheme that synthesizes image background so that models can achieve better generalization by learning from one synthetic image and generalizing to real natural images. The process involves using previous images as background and adding adversarial noises. Experimental results show that it can generalize from synthetic traffic signs and digit images to real traffic sign images and handwritten digit images, outperforming a baseline that directly trains from the synthetic images without augmentations or with random background augmentations.

**Main Review:**

-------------------------------------------------------
### Strengths

- **Problem setup:** Recognizing new concepts with one synthetic image is an interesting problem to study.

- **Good performance:** The proposed method achieves good performance on the proposed benchmarks.

- **Ablations:** The set of baselines is helpful for studying the effect of each component of the algorithm.


-------------------------------------------------------
### Weaknesses

- **Clarity:** It would be good to do some reorganization to make the paper clearer. Table-1 shows up the first but the description does not show up until section 5.3, and I was searching for the definition for bias correction. It seems like this term should show up in the model section too for defining what is bias correction. It is also not clear what does the “random initialization” or “bias heuristic” mean in Figure 3, and what is the horizontal axis.

- **Additional assumption:** The model assumes the knowledge of how to separate foreground and background in an image and explicitly synthesize augmented background to make it context-agnostic. Such information may not be available to some data and will require additional segmentation labels.

- **Lack of literature review:** There are some papers in the computer vision literature that study the contextual effect for classification, detection and segmentation, that can be probably mentioned. E.g.
[1] Shetty et al. Not Using the Car to See the Sidewalk -- Quantifying and Controlling the Effects of Context in Classification and Segmentation. CVPR 2019.
[2] Ntavelis et al. SESAME: Semantic Editing of Scenes by Adding, Manipulating or Erasing Objects. ECCV 2020.

- **More realistic benchmarks:** Currently, all the training images are synthetic. I wonder if such a method can be applied to more realistic benchmarks, such as the WILDS benchmark (Koh et al., 2021)?

- **Effect of representation pretraining:** It would be good to study a baseline that pretrains its representation from a large dataset such as ImageNet and then performs a transfer learning in this one-shot setting. Such a paradigm is very commonly used in few-shot learning these days. It can provide a useful representation to start with, instead of training from scratch from the one-shot dataset, which might lead to degeneracy.

- **IID test accuracy:** One drawback of the current design of the baselines is that the number of examples is perhaps too few for the network to learn a good hypothesis for the original distribution, let alone to learn a context-agnostic hypothesis for the new data distribution. So for example, what if the test set is not realistic images, but other traffic sign synthetic images with different transformations of the foreground only? Does the baseline perform well on these test images with just a blank background?

- **Upper bound performance:** What would be a performance upper bound, of using all the natural images to train the model? It would be good to set up such an experiment to understand the gap in between.

- **Random natural image background:** Instead of using the proposed procedure, another baseline to test is to use a random natural image (e.g. from ImageNet) as background. Although it uses more data, it is a very practical baseline method that can be easily adopted.

-------------------------------------------------------
### Questions for clarification

- **Foreground augmentation:** In Figure 1, it seems that the full model samples from the “object space” in step 1, which performs a certain level of foreground augmentation. Is this procedure also included in the baseline? If not, which of the baselines in Table 1 includes foreground augmentation? This seems like a very useful procedure and may introduce significant improvement over just training on one sample. This comment, combined with the previous comment, is trying to clarify the individual contribution of the context augmentation and foreground augmentation, and whether it is merely a question of the number of examples. If the “random context” does not have foreground augmentation, then it suggests that the foreground augmentation is playing a bigger role here. If it does have foreground augmentation, then I would suggest revising the text to make the augmentation procedure clearer.

- **Baseline vs. random:** It seems that both baseline and random use random background. Then what are the main differences between the two and what is the purpose of the random baseline?

-------------------------------------------------------
### Conclusion

In conclusion, this paper studies an interesting problem and proposes a simple and working method. However, since it compares to only baselines it creates and evaluates on synthetic benchmarks it proposes, I have to apply some extra scrutiny here, to make sure that the proposed method is widely applicable and robust, instead of just relying on some strict assumptions. In the above text, I mentioned a few more experiments and baselines that will help getting a more thorough picture on the empirical side. In addition, the clarity and literature review parts of the paper can also be improved.

-------------------------------------------------------
### Update post-rebuttal
I thank the authors for their response to my comments. The rebuttal has addressed some of my concerns regarding iid accuracy and the foreground augmentation, and therefore I have increased my rating from 5 to 6.

**Time Spent Reviewing:**

6

---

> ### Author Response · Authors · 2021-08-10
> **Author response**
>
> We thank the reviewer for taking the time to review our work and provide such thorough comments. We have responded to some of the major points below:
>
> > Clarity: It would be good to do some reorganization to make the paper clearer. Table-1 shows up the first but the description does not show up until section 5.3, and I was searching for the definition for bias correction. It seems like this term should show up in the model section too for defining what is bias correction. It is also not clear what does the “random initialization” or “bias heuristic” mean in Figure 3, and what is the horizontal axis.
>
> Thanks for bringing this to our attention--we will reorganize to improve clarity, but here are the answers to your questions:
> Random initialization refers to images whose backgrounds are initialized to random noise.
> Bias heuristic refers to images whose backgrounds are initialized to the previous image (in this case, the previous batch from the test set).
> The horizontal axis is the perturbation budget of the PGD adversary used to create the adversarial backgrounds (Section 4.2, Local refinement via robustness training). A budget of 0/255 means that we effectively do not perturb the background from the initialization, whereas a budget of 255/255 means that the background can be changed arbitrarily from initialization. Note that because the PGD method is a local search method, the initialization makes a difference even at 255/255.
>
> > Additional assumption: The model assumes the knowledge of how to separate foreground and background in an image and explicitly synthesize augmented background to make it context-agnostic. Such information may not be available to some data and will require additional segmentation labels.
>
> We have focused here on experiments which can validate our framework and theory, which requires the ability to sample from the object space and context space independently. The extensions suggested are interesting but outside our current scope.
>
> > Lack of literature review: There are some papers in the computer vision literature that study the contextual effect for classification, detection and segmentation, that can be probably mentioned. E.g. [1] Shetty et al. Not Using the Car to See the Sidewalk -- Quantifying and Controlling the Effects of Context in Classification and Segmentation. CVPR 2019. [2] Ntavelis et al. SESAME: Semantic Editing of Scenes by Adding, Manipulating or Erasing Objects. ECCV 2020.
>
> Thanks for suggesting these! We will be sure to include a brief discussion of these works. Note that both these works fall in the category of using generative networks to create or otherwise modify the context, whereas we hypothesize that one should not need any contextual information at all.
>
> > IID test accuracy: One drawback of the current design of the baselines is that the number of examples is perhaps too few for the network to learn a good hypothesis for the original distribution, let alone to learn a context-agnostic hypothesis for the new data distribution. So for example, what if the test set is not realistic images, but other traffic sign synthetic images with different transformations of the foreground only? Does the baseline perform well on these test images with just a blank background?
>
> Our models achieve nearly 100% accuracy with a (transformed, e.g., randomly rotated and scaled) synthetic traffic sign as foreground, with (1) a random background (Figure 3a, $\epsilon=0/255$) and (2) a previous test image used as the background (Figure 3b, $\epsilon=0/255$). These results suggest that our models do in fact learn a good hypothesis for the original distribution of synthetic traffic signs.
>
> > Upper bound performance: What would be a performance upper bound, of using all the natural images to train the model? It would be good to set up such an experiment to understand the gap in between.
>
> MNIST it is possible to get close to 100\%. Similarly, GTSRB can get around 99\%. The appendix has some additional comparisons to few-shot learning and domain adaptation approaches. In general, our method is very competitive with existing techniques, while using a completely different approach (and less data).
>
> > Foreground augmentation: In Figure 1, it seems that the full model samples from the “object space” in step 1, which performs a certain level of foreground augmentation. Is this procedure also included in the baseline? If not, which of the baselines in Table 1 includes foreground augmentation? This seems like a very useful procedure and may introduce significant improvement over just training on one sample. This comment, combined with the previous comment, is trying to clarify the individual contribution of the context augmentation and foreground augmentation, and whether it is merely a question of the number of examples. If the “random context” does not have foreground augmentation, then it suggests that the foreground augmentation is playing a bigger role here. If it does have foreground augmentation, then I would suggest revising the text to make the augmentation procedure clearer.
>
> The reviewer is correct. The foreground augmentation is present in all the methods presented (baseline, random-context, refinement-only, bias-heuristic, full), since we consider it part of sampling from the object space. The ablation studies differ only in how the background (i.e., context) is being generated.
>
> > Baseline vs. random: It seems that both baseline and random use random background. Then what are the main differences between the two and what is the purpose of the random baseline?
>
> The baseline picks a fresh random background for each image. The random-context initializes a background randomly, and reuses it again for subsequent images. This allows us to test whether just reusing backgrounds is sufficient to boost the performance, or whether the bias heuristic (which takes the entire previous image as the next background) is an improvement.
>
> To summarize:
> * baseline: random background
> * random-context: take previous background as next background
> * bias heuristic: take previous image as next background

---

### Decision · Program_Chairs · 2021-09-27

**Decision:**

Accept (Poster)

**Comment:**

This paper proposes a data augmentation scheme that synthesizes image background so that models can achieve better generalization by learning from one synthetic image and generalizing to real natural images.

All reviewers recommended to accept.

Accept.